# Composites of Semi-Rigid Polyurethane Foams with Keratin Fibers Derived from Poultry Feathers and Flame Retardant Additives

**DOI:** 10.3390/polym12122943

**Published:** 2020-12-09

**Authors:** Krystyna Wrześniewska-Tosik, Joanna Ryszkowska, Tomasz Mik, Ewa Wesołowska, Tomasz Kowalewski, Michalina Pałczyńska, Kamila Sałasińska, Damian Walisiak, Anna Czajka

**Affiliations:** 1Łukasiewicz Research Network, Institute of Biopolymers and Chemical Fibres, ul Skłodowskiej-Curie 19/27, 90-570 Łód’z, Poland; zdp@ibwch.lodz.pl (T.M.); e.wesolowska@ibwch.lodz.pl (E.W.); t.kowalewski@ibwch.lodz.pl (T.K.); protein@ibwch.lodz.pl (M.P.); d.walisiak@ibwch.lodz.pl (D.W.); 2Faculty of Materials Science, Warsaw University of Technology, Woloska 141, 02-507 Warszawa, Poland; joanna.ryszkowska@pw.edu.pl (J.R.); anna.czajka2.dokt@pw.edu.pl (A.C.); 3Department of Chemical, Biological and Aerosol Hazards, Central Institute for Labor Protection, National Research Institute, Czerniakowska 16, 00-701 Warsaw, Poland; kasal@ciop.pl

**Keywords:** semi-rigid polyurethane foams (SRPUF), keratin flour (CF), flame retardants, combustibility

## Abstract

Semi-rigid composites of polyurethane foams (SRPUF) modified with the addition of keratin flour from poultry feathers and flame retardant additives were manufactured. Ten percent by mass of keratin fibers was added to the foams as well as halogen-free flame retardant additives such as Fyrol PNX, expandable graphite, metal oxides, in amounts such that their total mass did not exceed 15%. Thermal and mechanical properties were tested. Water absorption, dimensional stability, apparent density and flammability of produced foams were determined. It was found that the use of keratin fibers and flame retardant additives changes the foam synthesis process, changes their structure and properties as well as their combustion process. The addition of the filler made of keratin fibers significantly limits the amount of smoke generated during foam burning. The most favorable reduction of heat and smoke release rate was observed for foams with the addition of 10% keratin fibers and 10% expandable graphite. Systems of reducing combustibility of polyurethane foams using keratin fillers are a new solution on a global scale.

## 1. Introduction

During recent years there has been a sharp increase in the use of natural fillers in the production of polymer materials. This was mainly due to the low price of the raw material derived from biomass, good usable properties of the polymer materials produced and ecological aspects.

Polyurethanes (PUR) is a group of polymeric materials, among others with excellent insulating properties [1]. According to the policy principles of sustainable development, raw materials from renewable sources were introduced into the recipes of polyurethane materials.

Pro-ecological activities have become an important element in the development of the PUR plastics industry. They mainly include partial or complete replacement of polyols of petrochemical origin with their plant counterparts and the introduction of so-called bio-fillers into the foam recipes [2,3,4].

As a plant-based filler for the production of polyurethanes, there were applied among others: vegetable fibers [5], lignin [6,7], cellulose [8,9], wood flour [10] and by-products from the food industry [11]. 

Natural fibers, which are available from renewable sources, are an attractive alternative as a reinforcing filler for rigid polyurethane foams (RPUF). There are known examples of the RPUF production based on rapeseed oil with the addition of flax fibers [12], walnut shells and microcellulose [13], based on PUR castor oil filled with wood flour [14], hemp and wood fibers [15], as well as RPUF strengthened with powdered eggshells [16]. Rigid foams are mostly used as thermal and sound insulation. The biggest disadvantage of polyurethane foams significantly limiting their use is their flammability.

The introduction of fire retardants impedes the thermal decomposition and ignition of the polymer, creating a coating on its surface, additional bonds or decomposing with the release of non-flammable gases or taking a significant amount of heat during the decomposition. These phenomena create a barrier for the fire, flammable gases and oxygen from air, the combustion mechanism changes, the amount of flammable gases decreases, the flammability of products surface decreases, the layers absorb heat, non-flammable gases are released (e.g., water vapor), the inflammation time lengthens [17].

The halogen flame retardants used recently give off pungent and toxic smoke, which negatively affects the environment, is harmful for human health and life [18]. For this reason, the European Union has introduced restrictions on their use in the member states. Intensive studies are carried out to obtain materials with the best flame retardant properties that would not contain halogen fire retardants. For this purpose are mainly used: expanded graphite, phosphorus and nitrogen compounds, metal hydroxides and nanofillers. Phosphorus-containing compounds are an important group of flame retardants that have low environmental impact. They do not emit toxic gases during burning and are characterized by low toxicity.

For flame retardancy of polyurethane foam the following compounds are used: phosphates, red phosphorus, phosphites, phosphonates and alkyl phosphamides [19,20,21]. Ammonium polyphosphate (APP) is the most commonly used phosphorus flame retardant. Phosphorus both reduces flammability, as well as creates a glassy flame retardant layer on the surface of the material. In order to reduce flammability, a system of flame retardants is often introduced [22]. The use of sodium dihydrogen phosphate with sodium pyrophosphate gives a synergistic flame retardant effect of rigid polyurethane foams by forming a carbonized layer (phosphoric acid) and thermal stability (excess sodium dihydrogen phosphate). A large amount of nitrogen in the polymer structure reduces its flammability similar to phosphorus [22].

Melamine (C_3_N_3_(NH_2_)_3_) and urea (NH_2_CONH_2_) are the most commonly used nitrogen-containing flame retardants. During combustion, melamine absorbs the heat generated by the PUR matrix [23] and the melamine condensation products formed during heating form a charred layer on the polymer surface, limiting its flammability. The introduction of halogen-free flame retardants into the polyurethane matrix that is, (ammonium polyphosphate (PFA) and the system of ammonium polyphosphate and melamine cyanurate) [24], (poly (ammonium phosphate), triethylene phosphate and Fyrol-6) [25,26] caused deterioration mechanical properties of obtained systems but significant improvement of thermal stability.

Aluminum hydroxide (ATH) and magnesium hydroxide are the most commonly used inorganic fire retardants in the flame retardant technology of polymeric materials. These hydroxides in contact with fire decompose endothermically at 205 °C (ATH) and 300 °C Mg(OH)_2_, respectively. During decomposition water vapor is released, which after entering the combustion zone limits the concentration of flammable gases and oxygen. The oxides formed settle on the surface of the material, creating a protective layer that limits the transport of volatile products of the flame and oxygen into the material. It also reduces smoke emissions. There are known studies based on metal oxides and bimetallic oxides (Cu_2_O, NiO, MoO_3_, CuMoO_4_ and NiMoO_4_) in order to reduce toxic products produced under various conditions of thermal decomposition of nanocomposite polyurethane foams [27].

Aluminum hydroxide is used for polymers processed at temperatures up to 200–220 °C [28]. The introduction of ATH into rigid polyurethane-polyisocyanurate foams caused that these foams in the horizontal test of burning time were determined as self-extinguishing [29]. The content of flame retardant in the polymer generally reaches 40% by mass.

The addition of such amount of compound which is non-polymer has a significant impact on the physico-mechanical and processing properties of the material. Decreasing the content of flame retardant compounds can be achieved by the so-called synergistic effect. It is based on the fact that the total effect in delaying the combustion of a mixture made of two or more components may be greater than the sum of the individual actions of these components. Obtaining synergistic systems requires appropriate choice of ingredients [30,31,32].

The action of expanded graphite (EG) as a fire retardant consists in the formation of a charred layer [33], which acts as an insulator due to the formation of small air gaps between graphite layers. EG significantly reduces heat and weight loss, smoke production and the emission of toxic fumes. Not all forms of expanded graphite can be used to reduce the flammability of plastics. Low-temperature expanded graphite is used as the flame retardant. Swelling occurs when the so-called critical temperature is achieved. It is the temperature at which exothermic reaction, decomposition and ignition occur spontaneously. The charred layer forms a thermal barrier that limits oxygen diffusion and prevents later degradation of the polymer matrix [33]. The use of expanded graphite as a flammability modifier for rigid polyurethane foams has been extensively described in the literature [34,35,36,37,38].

Interesting fire retardant are grounded poultry feathers [39]. Waste protein materials of the keratin type have aroused the interest of scientists for many years [34]. They are attractive not only for medicine and biotechnology but also as a component of composite materials, characterized by, among others, their barrier properties. The use of this waste group has a significant impact on the sustainable development of the state, as it creates an opportunity to increase the amount of resources used from renewable raw materials.

One of the least known and undeveloped is waste in the form of poultry feathers, extremely rich in keratin (about 95%). Every year, millions of tons of this waste are generated all over the world [35], which on one hand means huge environmental pollution and on the other hand gives an inexhaustible source of valuable protein. Waste in the form of poultry feathers consists mainly of keratin, which has a hydrophobic character. The multilevel structure of keratin is maintained by cystine disulfide bridges and hydrogen bonds. Such structure is the reason for the high resistance of feather keratin to chemical and physical agents [40].

Thanks to their advantages, these fibers can be successfully used as a filler for composite materials. They are cheap material and available in huge amount, it is characterized by low density (0.89 g/cm^3^) compared to traditional cellulose fillers. They are excellent insulators both thermal (they show low heat conductivity) and acoustic [39]. These properties result from the porous structure of the fibers filled with air [41].

A very important feature is the structure of keratin fibers, containing both an amorphous and crystalline phase, increasing mechanical strength and ensuring the high Young’s modulus of polymer composites filled with poultry feather fibers [42,43,44].

The purpose of the work was to use waste poultry feathers as a bio-filler for semi-rigid polyurethane foams, then to assess their properties and indirectly the usefulness of this filler in the manufacture of polyurethane plastics.

## 2. Experimental Part

### 2.1. Materials

Semi-rigid foams (SRPUF) were made using a mixture of the following substrates:Arcol^®^Polyol 1374, a trifunctional polyetherol with a hydroxyl number LOH = 26, water content below 0.1% by mass; (Bayer, Bergkamen, Germany),Daltocel F526—polyetherol with LOH = 128 hydroxyl number, (Huntsman Corporation, The Woodlands, TX, USA),Diisocyanate Ongronate 4040, a mixture of monomeric isomers and oligomeric methylenediphenyl-4,4’-diisocyanate (MDI); (BorsodChem, Kazincbarcika, Hungary),Distilled water.


The foams were made using the following flame retardant additives:
Keratin filler (sulphur content of 2.9%, nitrogen content of 15.5% and ash content of about 1%) with particle size 0.01–0.04 mm (K), (Łukasiewicz Research Network—Institute of Biopolymers and Chemical Fibres, Łód’z, Poland), aspect ratio of fibers = 2.59, the SEM image of the fibers is shown in Figure 1.Fyrol PNX (F)—(ICL Industrial Products Ltd., Tel-Aviv, Israel), oligomeric non-reactive phosphate esterExpandable Graphite (GE)—(Sinograf SA, Toruń, Poland), the particle size 0.5 mm, expansion 250 mL/g starting expansion temperature 220 °CAluminum hydroxide, MARTINAL (ATH)—(Albemarle, Charlotte, NC, USA), the particle sizes 10 µmMagnesium hydroxide, MAGNIFIN (MTH)—(Albemarle, Charlotte, NC, USA), the particle sizes 20 µmZinc oxide (ZO)—Institute of High Pressure Physics (Unipress, Warsaw, Poland), the particle sizes 70 nmAmmonium polyphosphate, Exolit AP 422, (APP)—(Clariant, Muttenz, Switzerland).

### 2.2. Preparation of Foams

The foams were made by one-step method. Flame-reducing additives were introduced into the polyol component (the amount of additives was calculated per 100 g polyol). Foam synthesis was carried out at an ambient temperature of 20 °C. Polyols and modifying additives (polyol master batch) were mixed with a high speed stirrer at 800 rpm for 60 s.

Then the filler was introduced and the whole system was mixed with the use of a mechanical stirrer at 200 rpm for 30 s. The foams were made at the isocyanate index of 110. The description of the foam composition and synthesis process parameters is presented in Table 1.

## 3. Research Methodology

Obtained rigid polyurethane foams were characterized by infrared spectroscopy, thermogravimetric analysis and scanning electron microscopy. Apparent density, flame resistance, heat conduction coefficient, moisture absorption were determined as well as thermal parameters using a cone calorimeter.

### 3.1. Determination of Apparent Density

Apparent density was determined on the basis of independent measurement of mass and volume of the sample. An electronic scale was used to determine the mass. Measurements were carried out at room temperature. The apparent density was calculated using Equation (1):(1)$$d=\;\frac{m}{V},$$
where *m* is sample mass [g], *V* is sample volume [cm^3^].

The mass of the samples was determined with an accuracy of ±0.001 g and the dimensions of the samples were measured with an accuracy of ±0.01 mm.

### 3.2. Description of the Chemical Constitution and Structure of the Foams (ATR-FTIR)

The chemical structure of the foams was assessed by attenuated total reflection Fourier-transform infrared spectroscopy (ATR-FTIR). The analysis was performed using a Nicolet 6700 spectrometer (Thermo Electron Corporation, Waltham, MA, USA) with an ATR (attenuated total reflection) attachment. Each sample was scanned 64 times in the 4000–400 cm^−1^ wave number range. Scan results were analyzed using Omnic Spectra 2.0 (Thermo Nicolet Spectrometer, Thermo Fisher Scientific, Waltham, MA, USA) software.

### 3.3. Differential Scanning Calorimetry (DSC)

The test was performed on DSC Q1000 calorimeter (TA Instruments Inc., New Castle, DE, USA) to describe thermal transformations taking place in foams. Each sample weighing 5 ± 0.2 mg was sealed in aluminum dishes. The measurement was automated. In the first stage of measurement, the samples were cooled to −90 °C and then heated to 210 °C at a rate of 10 °C/min in a helium atmosphere. In the next cycle, the samples were cooled and in the third, they were heated again. Foam parameters were determined on the basis of the first and third cycle. The obtained thermograms were analyzed in the Universal Instruments software version 4.7A (TA Instruments Inc., New Castle, DE, USA).

### 3.4. Thermal Degradation

Thermal degradation of the foams (TG) was analyzed using a TGA thermogravimeter (TA Instruments Inc., New Castle, DE, USA, model Q500). Tested samples weighing 10 ± 1 mg, placed in platinum dishes, were heated at a rate of 10 °C/min from room temperature to 1000 °C under a nitrogen atmosphere. The results were analyzed using Universal Analysis 2000 software version 4.7A (TA Instruments Inc., New Castle, DE, USA). The TGA study was conducted in a nitrogen and air atmosphere.

### 3.5. Flame Resistance Test

For the analysis of the rate of heat and smoke release by polyurethane foams a cone calorimeter (Fire Testing Technology Ltd., East Grinstead, UK) was used. Tests were carried out according to ISO 5660: 2002 using samples with dimensions 100 × 100 × 8 mm. The test was carried out at 30 kW/m^2^.

### 3.6. Heat Transfer Coefficient, λ

The thermal conductivity coefficient (λ) was determined using the FOX 200 Heat Flow Meter (TA Instruments Inc., New Castle, DE, USA) at temperature difference of hot and cold plate 20 °C. Measurements to compare the λ values of different foams were carried out at medium temperature 0 °C (cold plate temperature −10 °C, hot plate temperature 10°); 10 °C (cold plate temperature 0 °C, hot plate 20 °C) and 20 °C (temperature cold plate 10 °C, hot plate 30 °C). Samples with dimensions of 200 × 200 × 50 mm were prepared for this purpose. The measurement of the thermal conductivity coefficient was made after 14 days from the moment of foam synthesis.

The described method is based on determination the amount of heat flowing through the sample in a unit of time, while the heat flow is determined at a constant temperature difference on opposite sides of the tested material. The value of the thermal conductivity coefficient is determined from the Fourier equation: (2)$$q=\;-\lambda \frac{dT}{dx},$$
where *q* is the density of total heat flux (W/m^2^) transported on the road *x*, *λ* is the thermal conductivity coefficient, (W/m·K), dT/dx is the temperature gradient in the x direction, (K/m).

### 3.7. Water Absorption Test

The test was carried out according to the Standard PN-93/C-89084. Three samples were cut from the foams, their dimensions were determined and weighed. The samples stayed in the water for 24 h, after which time they were weighed again and their dimensions were measured one week after exposure to water. The water absorption was determined on the basis of the following relationship:(3)$$Ch_{H_{2}O}=\frac{\frac{m_{2}-m_{1}}{\rho_{w}}}{V_{1}}*100\%,$$
where *m*_2_ is the sample mass after 24-h exposure to water [kg], *m*_1_ is the dry sample weight before the test [kg], *ρ_w_* is the water density at 25 °C (997 kg/m^3^), *V*_1_ is the volume of dry foam [m^3^].

After one week the foams were again measured to verify dimensional stability. The change of dimensions is described by the relationship:(4)$$S_{w}=\frac{V_{1}}{V_{2}}*100\%,$$
where *V*_1_ is the volume of dry foam before the test [m^3^], *V*_2_ is the foam volume one week after being removed from water [m^3^].

### 3.8. SEM Analysis

The microstructure of the keratin fibers, reference foam and composites was analyzed using Scanning Electron Microscopy (SEM) TM3000 Hitachi High-Tech Corporation, Tokyo, Japan). Samples were sputtered using gold-pallad target of the sputter coater Polaron SC7640 (Perkin Elmer, Waltham, MA, USA). The sputtering was carried out under 6 mA current intensity and 2 kV voltage for 80 s. Five types of foams and composites were observed. The observations were made using 5 keV acceleration voltage. The mean micropore diameter (d) was determined by delimiting the area of 120 micropores. The mean aspect ratio (AR) was determined based on the formula:(5)$$AR=\frac{d_{max}}{d_{min}},$$
where *d_max_* and *d_min_* are the major and minor axis of the pore [µm]. *d_max_* and *d_min_* were determined using by delimiting the area of 120 micropores. Delimiting the pores area were performed using ImageJ software. Observations of foam and composites were performed in the perpendicular to the direction of growth.

## 4. Results and Discussion

Within the work, semi-rigid polyurethane foams (SRPUR) were manufactured containing 10% of keratin fibers and compounds that increase their resistance to fire.

Among the inorganic compounds, aluminum hydroxide (ATH) and magnesium (MTH) and zinc oxide (ZO) were selected. Fyrol PNX (F) was used from the group of organophosphorus compounds and ammonium polyphosphate (APP) from the group of nitrogen and phosphorus compounds. Expandable Graphite (GE) was also used. During the synthesis of foams, their course was analyzed (Table 1). The start time of the foams was very short, less than 10 s, therefore only the growth time and gelation time were recorded for the tested foams. Growth time of foams without modification was 75 s and modified foams in the range of 49–73 s. The introduction of keratin fibers reduces the growth time of foams by approx. 15%. The growth time for foams with the addition of ATH and MTH varies around 64 s, the growth time of foams after the ZO addition is shortened by approx. 30%. Growth time is also shortened after application GE.

Gelation time of the initial foam is 88 s and for other foams it ranges from 72 to 94 s. There is observed significant shortening the gelation time of foams after the introduction of ZO, by approx. 17%. Changes in the course of foam synthesis are the result of changes in pH caused by the introduction of various additives. Decreasing the pH leads to an increase of the reaction rate of isocyanate groups with both hydroxyl, urethane and urea groups [45].

To analyze changes in the chemical constitution of foams containing keratin fibers and Fyrol, ATR-FTIR analysis was performed (Figure 2).

Presented spectra confirmed the presence of chemical groups for rigid polyurethane foam samples. Signal at 3345 cm^−1^ corresponds stretching, symmetrical and asymmetrical vibrations, assigned to the N-H bond [46]. The clearly visible signals at 2867 cm^−1^ and 2970 cm^−1^ comes from symmetrical and asymmetrical stretching vibrations within the C-H_2_ groups in soft segments arising from polyols [47]. Signals around the 2270 cm^−1^ attributable to -NCO bond from unreacted isocyanate were observed [48].

In all analyzed samples there were observed signals originating from the stretching vibrations of C=O bonds (1709 cm^−1^), C=C from the aromatic ring (1595 cm^−1^), bending and deformation vibrations originating from NH bond within -NHC = O (1538 and 1511 cm^−1^), CCH_3_ (1458 cm^−1^), -O-CH_2_ (1413 cm^−1^) or CO ν asymmetrical/symmetrical within the group -N-CO-O (1222 and 925 cm^−1^) [46].

Absorption around 762 cm^−1^ represents a C-H bond derived from an aromatic ring. The introduction of keratin into the sample does not cause significant changes in the intensity of the signals in the foams. After introduction of Fyrol PNX the clear changes in intensity and shifts of peaks were observed. The intensity of the peaks increases at, 1222 cm^−1^, 1090 cm^−1^, 1015 cm^−1^, 812cm^−1^, 762 cm^−1^ and 697 cm^−1^.

The increase in the intensity of the amide peak III—1222 cm^−1^ indicates that the use of a mixture of keratin and Fyrol catalyzes the reaction of the -O-H with the NCO groups and it is confirmed by the observations made on the basis of the analysis of the synthesis process. Phase structure of the foams was also thermally analyzed using DSC. Examples of DSC thermograms of the analyzed materials are presented in Figure 3 and Figure 4 and the results of their analysis in Table 2.

On the base of DSC thermograms, glass transition temperature of the soft phase was determined in the first heating cycle (Tg_1_) and in the second heating cycle (Tg_2_). In addition, in some of the DSC curves from the second heating cycle, the glass transition temperature of hard phase (Tg_3_) was observed and in some curves the second glass transition temperature in the hard phase area (Tg_4_) is also marked.

On some curves obtained during the first heating cycle an endothermic peak with a minimum at temperature T and the enthalpy of transformation ∆H appears.

The introduction of inorganic fillers as flame retardants causes slight changes in the thermal characteristics of foams. Comparison of DSC thermograms of a series of selected foams is presented in Figure 4.

Temperature of glass transition of the soft phase determined in the first soaking cycle reaches from −60.5 to −62 °C and in the second heating cycle from −61.8 to −64.6 °C. The increase of the glass transition temperature of the soft phase after the heating process indicates an increase in the degree of phase separation in the foams [49]. In addition, during the second heating cycle, the endothermic transformation observed during the first heating cycle does not appear. This transformation is related to change of arrangement in the hard phase of foams, which occurs in the temperature range of 40–130 °C with a minimum at 75–85 °C and the enthalpy of this transformation is 31.4–41.5 J/g. Description of this transformation was based on the results of studies regarding polyurethane elastomers that were presented in papers [50,51,52]. This transformation occurs in such a wide temperature range, because it is the result of changes in arrangement of the interface, which is a mixture of flexible segments and rigid segments with different chemical structure [1,2]. The characteristics of this transformation depends on the type of additives used for modification of the foams. In the second heating cycle in all analyzed foams, the glass transition temperature Tg_3_ was observed in the temperature range 107–110 °C, in some foams there was also a second temperature Tg_4_ in the temperature range 148–152 °C. The first temperature is the result of the formation of a hard phase from rigid segments with greater flexibility than the second phase described by the second temperature. The analysis of phase structure can be indirectly described based on the foam decomposition process evaluated by TGA analysis in nitrogen. An example of the result of the TGA analysis for sample R is shown in Figure 5. On the base of the mass change curve (TG) the loss temperature of 2%, 5%, 10% and 50% as well as the degradation residue at 700 °C—U700 was determined (Table 3).

From the mass derivative curve (DTG), the maximum degradation rate temperature of individual stages of decomposition observed on these curves (T1, T2, … Tn) and the maximum rate of degradation of these stages (V1, V2, … Vn) were determined. The temperature range in which each of the stages occurs was also determined, as well as the weight loss during each of the stages. An example of determining this data is shown in Figure 5.

Comparison of selected DTG curves of foams analyzed in nitrogen atmosphere is shown in Figure 6 and the results of the analysis are listed in Table 3 and Table 4.

Based on the analysis of Figure 6, it can be concluded that the introduction of keratin and Fyrol changes the foam degradation process. There are two degradation stages in R foam while three stages in R+10K and R+10K+5F foams. After the introduction of keratin, the beginning of the degradation process is slower (the first and second stages related to degradation in the range of hard phase) and the components of the soft phase decompose faster. The introduction of Fyrol (R+10K+5F) accelerates the beginning of the degradation process and this process begins at lower temperatures.

T2% and T5% slightly decreases after using keratin and clearly decreases after adding to the foam with keratin additionally F, ATH, MTH, ZO, GE and mixtures of F and APP even by about 30 °C. Whereas T10% for all foams varies slightly on the level of 283 ± 8 °C. It was observed that the weight loss by 50% for foams with APP occurs at similar temperatures to foam R, while for other foams it occurs at a much higher temperature even by 12 ± 21 °C. The introduction of keratin does not cause significant changes in the amount of residue (U) after the degradation process at 700 °C [53,54]. The amount of residue during degradation process at this temperature increases when other additives and their mixture with Fyrol are used.

The addition of Fyrol into foams causes that degradation process starts at lower temperatures. This stage takes place in the temperature range 160–220 °C. During this stage the sample mass decreases by approx. 2%. At this stage there may be degradation of biuret or allophanate groups, which may have been formed during a slower process.

The second stage of degradation occurs at temperatures around 220–280 °C. In this stage there is a noted sample weight loss from 7% to 12% associated with the degradation of urethane and urea groups in the hard phase.

The third stage occurs in the range of about 280–330 °C. During this stage, about 7–19% of the sample mass decreases. At this stage the ether bonds are degraded in the soft phase.

The fourth stage is related to the degradation of decomposed products from previous stages.

The thermal degradation process of tested foams in the air atmosphere was examined using TGA analysis and the results are summarized in Table 5 and Table 6.

In the analyzed foams the temperature differentiation is noted at which 2% weight loss occurs. Compared to the reference foam, the introduction of some fillers significantly increases the temperature at which 2% weight loss occurs, these are: keratin, a mixture of keratin and Fyrol, as well as a mixture of keratin and Fyrol with the addition of ATH, MTH, moreover, ZO, GE and APP cause an increase in T2%.

A 5% weight loss occurs in all foams at a similar temperature of 232–241 °C, similarly in case of 10% weight loss (242–249 °C).

The introduction of additives causes a significant temperature increase at 50% weight loss. The highest temperature at 50% weight loss was observed for APP foams (increase by approx. 75 °C).

For foams analyzed in the atmosphere of air the 2 or 4 degradation stages are observed. The first stage of degradation occurs at a temperature of about 220–270 °C and the second stage in the temperature range of 270–310 °C, in the case of some foam a degradation stage also appears in the range of 306–390 °C.

In total, during these stages there is a loss of about 50–57%. During the last stage of degradation, there is a decomposition of products which did not decompose during the previous stages, aromatic bonds and other products, as well as ether bonds. It can be supposed that the first stage is related to the degradation of urethane bonds, the second degradation stage is attributed to urea bonds and the third one to disubstituted urea bonds. During the first stage of degradation in air, decomposition takes place at the highest rate of approx. 0.94–1.92%/°C. The lowest decomposition rate was observed for the following samples: R+10K+10GE+5F; R+10K+10APP; R+10K+10APP+5F.

The use of Fyrol cause the increase the decomposition rate when introduced into the mixture with ATH, MTH and ZO by 2, 12 and 6%, respectively. The speed of the second and fourth degradation stage was three times lower.

Figure 7 summarizes the heat release rate (HRR) curves determined for the SRPUF during the test carried out with the cone calorimeter. A comparison of the effect of the addition of keratin and various types of flame retardants on the change of the HRR curve shows that keratin and selected systems containing keratin cause a decrease in pHRR and a slight flattening of the curve, while the others are quite the opposite.

For example, the combination of keratin and aluminum hydroxide slightly decreased the pHRR value compared to the unmodified foam but this value was higher than that obtained for R+10K (Figure 7a). The best results from this series were obtained for the foam modified with a flame retardant system containing keratin, ATH and Fyrol, which is also characterized by the highest proportion of flame retardants.

Similar relationships were observed for the series with magnesium hydroxide (Figure 7b), although the keratin, Fyrol and MTH system did not bring such a large effect in reducing pHRR as the system with ATH. On the other hand, the introduction of zinc oxide resulted in a significant increase in the HRR value, despite the increase in the share of flame retardants, which suggests an antagonistic effect (Figure 7c).

The best effects in limiting the heat release rate were noted for samples containing expandable graphite. The curves shown in Figure 7d suggest that GE caused the creation of a protective layer that effectively limited the access of the heat flux to the deeper layers of the material. It should be noted that the addition of Fyrol caused a further reduction in HRR. Examples of the occurrence of a synergistic effect between expanded graphite and phosphorus flame retardant as well as layered aluminosilicates, leading to a limitation of the flammability of polyisocyanurate foams, have been described earlier in the literature [55]. The use of APP influenced the course of the HRR curve but the obtained results were not as favorable as in the case of the series with ATH, MTH or GE.

Table 7 summarizes the key parameters determined on the basis of combustion tests carried out with the use of a cone calorimeter, that is, time to ignition (TTI), end time of flame combustion (TTF), maximum heat release rate (pHRR), maximum average heat emission factor (MAHRE), total heat released (THR), mass loss rate (MLR) as well as specific extinction area (SEA) and the total amount of smoke produced (TSP). In addition, the pHRR ratio to time of its achievement was calculated, which, similarly to the MARHE parameter, provides information about the possibility for fire growth and spread [56].

A significant extension of the time after which the samples became inflamed (TTI) was observed in the case of the K+ATH+F and K+MTH+F systems, which is connected with their endothermic decomposition process and the release of water (ATH loses water at 220–250 °C, whereas MTH at 300–340 °C), which makes the burning process more difficult [57]. On the other hand, keratin itself, keratin with Fyrol and both components in combination with APP resulted in a shorter TTI. All samples burned longer than unmodified SRPUF and the longest flame burning time (TTF) was observed for samples containing graphite.

The use of keratin decreased the pHRR value by about 21% but the additional introduction of Fyrol contributed to the increase of the value of the tested parameter to the level specified for the unmodified SRPUF. The greatest reduction in pHRR was achieved for foams with keratin and GE (decrease by 54%) and with keratin, GE and Fyrol (decrease by 59%). This is probably due to the barrier effect of the formed char, reducing the permeation of oxygen and the escape of volatile degradation products [58]. Moreover, for both materials, the MARHE parameter decreased more than twice. Nevertheless, for the R+10K+10GE sample, the highest value of the pHRR ratio to the time of its achievement was also obtained, which was the result of the maximum HRR value occurring shortly after the sample ignited (Figure 7g).

The introduction of the analyzed substances increased the total heat released, the highest in the case of the R+10K+ZO sample. Similarly, for all materials, an increase in the rate of weight loss was observed and the values closest to those determined for unmodified foam were obtained for the GE samples.

In the case of all foams modified with flame-retardant additives, a decrease in SEA was observed. This index decreased by 72% after the use of keratin alone and a further reduction in SEA was observed after the addition of one of the additives such as Fyrol, aluminum hydroxide and expanded graphite (by 80%, 84%, 94%, respectively). The use of a system consisting of 3 substances caused the increase SEA, even 6-fold in the case of R+10K+10GE+5F foam. The only exception was the 10K+10APP+5F system. In turn, for the total amount of smoke released, a decrease in the value of the analyzed parameter was observed only for materials marked as R+10K+10ATH; R+10K+10APP+5F and R+10K+10GE and the reduction was 8%, 41% and 61%, respectively. The smoke suppression effect depends on the mechanism of each fire retardants used and it is caused by dilution of the combustion zone and/or by a barrier effect of the formed char. ATH decomposes endothermically under the heat radiation, causing water released and dilutes the combustible gases and alumina forms insulation on the surface of the burning polymer [57].

Non-thermal hazards, namely toxic smoke, cause the predominant cause of death for fire victims [59]. Fang et al. [60] reported that victims would suffocate when the concentration of CO and CO_2_ reaches 1% and 5%, respectively. As can be seen in Table 7 the addition of keratin affects slightly on the growth of peak of CO emission (pCO) and CO_2_ emission (pCO_2_), while, Fyrol doubles the pCO value and reduces pCO_2_. Similarly, the combination of Fyrol with ATH, MTH and ZO in most cases caused an increase in pCO and decrease in pCO_2_. Fyrol PNX, as a fire retardant with a high phosphorus content, works mainly in the gas phase but a small amount also remains in the condensed phase [61]. Therefore, for fire retardants acted in the condensed phase and exhibiting the ability to form char (GE, APP), a reduction in both parameters after Fyrol addition can be observed. In the literature, we can find that the use of ammonium polyphosphate in polyurethane foams may pointedly reduce the emission of CO, smoke density and the formation of soot similar to expandable graphite who limits toxic gases although in a lower proportion than APP [57]. Probably, the incorporation of the major products of incomplete and complete combustion into the structure of the char reduced its amount in emitted gases.

In the case of some analyzed materials, the influence of burning time on the increase in THR and TSP values cannot be ruled out. Taking into account both the intensity of the combustion process and the emission of smoke, the most favorable results were obtained for the foam containing keratin and expanded graphite. GE works mainly in the solid phase, creating a layer of low-density char on the surface of the polymer but it also greatly reduces smoke. This is caused by the reaction of sulfuric acid with graphite flakes, which expands its volume by about 100 times, accompanied by gases generated at high temperature (CO_2_, H_2_O, SO_2_) [62].

Thermal conductivity test was carried out for foams containing expandable graphite. Foam R has a thermal conductivity coefficient λ = 0.036 W/(m·K)), while foam with graphite R+10K+10GE and R+10K+10GE+5F this coefficient is 0.039 W/(m·K) [63].

Dimensional stability and water absorption were determined for the foams manufactured and the results are summarized in Table 8.

The introduction of keratin has effect on increase the foam dimensional stability and decrease water absorption [64]. Other foams modifications result in greater changes in foam dimensions and higher water absorption than R foam.

To clarify whether the cell structure of the foams affects their properties, SEM images of the reference foam and selected composites were made. The images of the foams are shown in Figure 8.

The introduction of keratin fibers causes a significant change in the pore size of the foams. In the samples R+10K (Figure 8b) and R+10K+5F (Figure 8c), the appearance of a larger number of pores with smaller sizes was observed compared to the sample R (Figure 8a). In Figure 8b,c no differences due to Fyrol incorporation were observed. The incorporation of GE into the R+10K foam (Figure 8b) causes the R+10K+10GE foam (Figure 8d) to produce more large-sized pores. A similar change was observed after the introduction of GE into the R+10K+5F foam (Figure 8c), more large-sized pores are formed in the R+10K+10GE+5F foam Figure 8e). The observation of the GE structure in the R+10 K+10GE+5F foam (Figure 8f) indicates that the flake structure of the agglomerates of this filler was preserved. Figure 8g shows a single keratin fiber located in the foam R+10K season.

The quantitative analysis of the images of the foam pore structure was performed, the results of which are summarized in Table 9. The highest mean pore diameter (d) was observed in reference foam sample. The lowest was observed in R+10K and R+10K+5F samples. The introduction of keratin fibers into the R reference foam reduces the d of R+10K and R+10K+5F composites. It was found that in the composite with Fyrol (Figure 8c) the d distribution was greater than in the R+10K foam (Figure 8b). The aspect ratio (AR) of the pore ovals of the keratin fiber foams (Figure 8b,c) indicates that it is closer to the oval than the reference foam (Figure 8a). The introduction of GE to keratin foams (R+10K+10GE and R+10K+10GE+5F) increases the d in comparison to R+10K foams. In the R+10K+10GE+5F foam, the use of Fyrol causes a significant increase in the size distribution of the d but the AR of the pores of this foam indicates that their shape is closer to the oval shape than in the R+10K+10GE foam.

Fyrol foams are characterized by a much higher water absorption, which may be related to the increased spread of the AR of the foams after its introduction. In foams with Fyrol, more large-sized pores are created, which may reduce the burning time of the foams (TTF).

## 5. Conclusions

The study was designed towards the manufacture of a series of foams containing 10% keratin which were modified with various groups of flame retardant additives. These foams were characterized, their phase structure, thermal degradation and flame resistance were investigated. Their dimensional stability and water absorption were also examined.

Experimental results indicate that application of keratin significantly reduces the amount of smoke generated during foam burning. Fyrol increases the ignition time, decreases the maximum heat release rate when used in systems with other additives but increases the amount of smoke emitted during burning. The most advantageous features in contact with the flame were observed for foams modified with the mixture of keratin, graphite and Fyrol. These foams are also characterized by a little change in thermal conductivity and slightly higher water absorption.

## Figures and Tables

**Figure 1 polymers-12-02943-f001:**
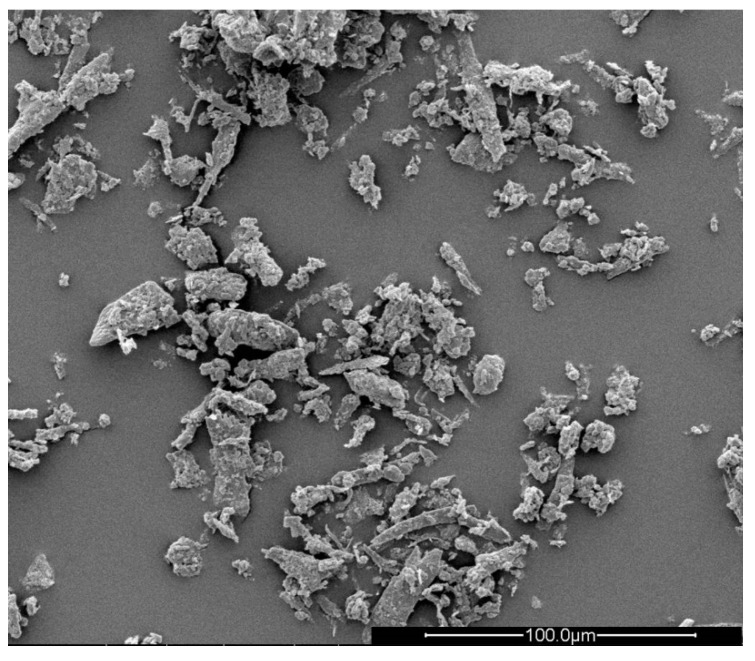
Scanning electron microscopy (SEM) image of keratin fibers.

**Figure 2 polymers-12-02943-f002:**
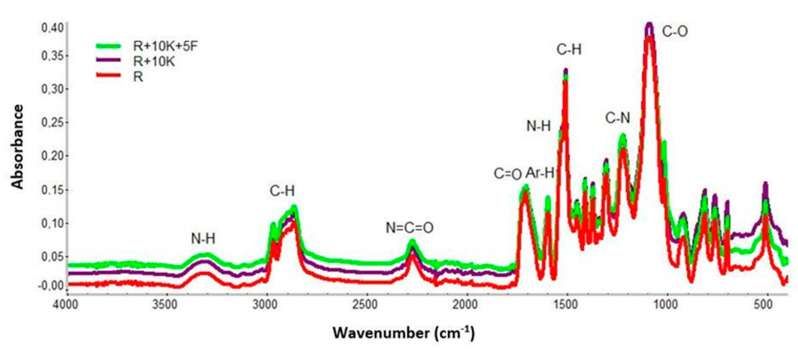
Comparison of Fourier transform infrared (FTIR) spectra of samples modified with keratin and Fyrol PNX.

**Figure 3 polymers-12-02943-f003:**
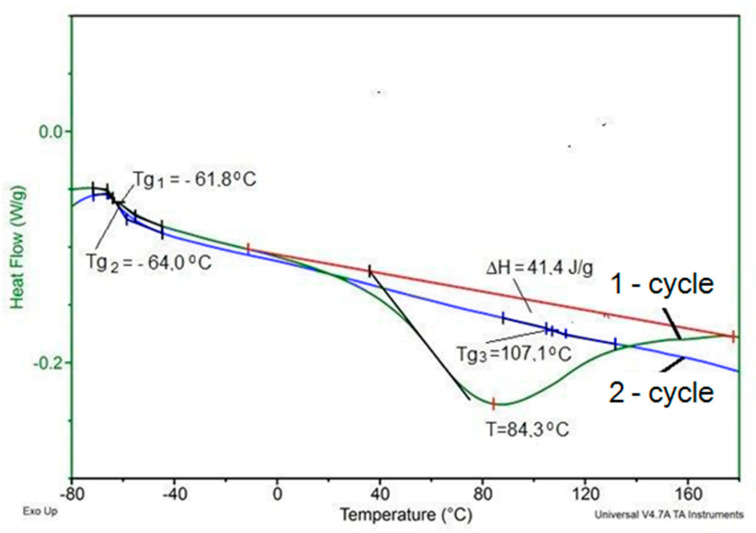
Example of differential scanning calorimetry (DSC) curve for sample R+10K+5F.

**Figure 4 polymers-12-02943-f004:**
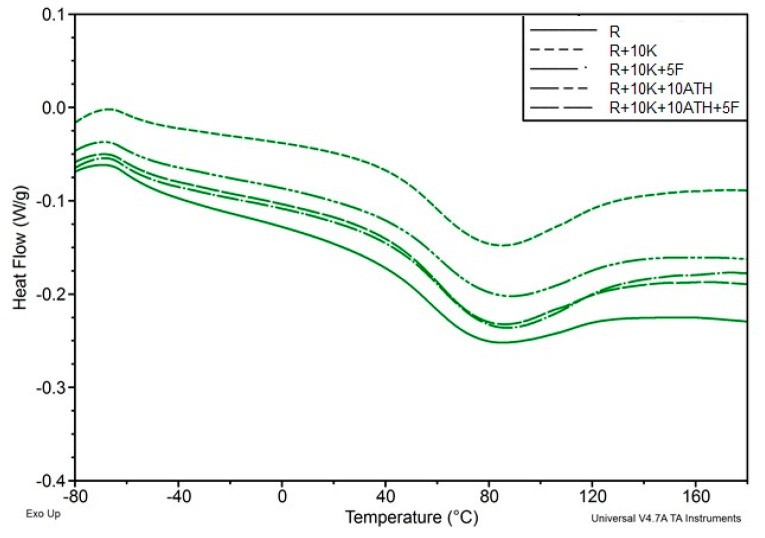
Comparison of DSC thermograms of foams with ATH, keratin and Fyrol with reference foam obtained during the first heating cycle.

**Figure 5 polymers-12-02943-f005:**
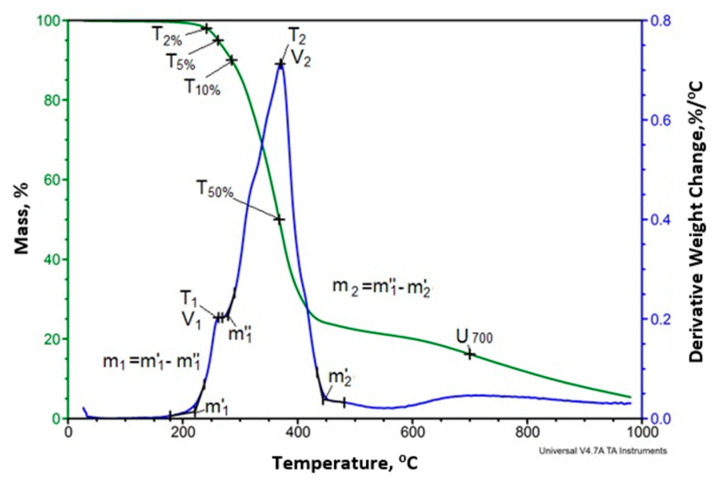
The examples of thermogravimetric analysis (TGA) curves for foam sample R.

**Figure 6 polymers-12-02943-f006:**
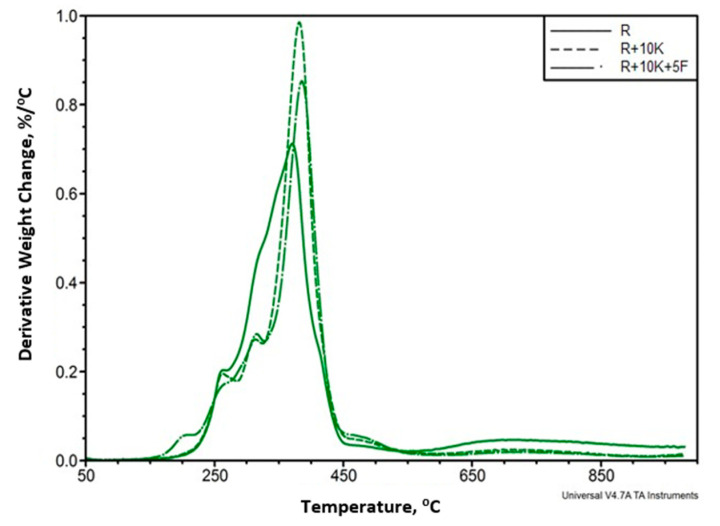
Summary of the results of the Derivative Thermogravimetry DTG analysis of the reference sample (R), foam containing 10% keratin (R+10K) and foam containing 10% keratin and 5% Fyrol (R+10K+5F).

**Figure 7 polymers-12-02943-f007:**
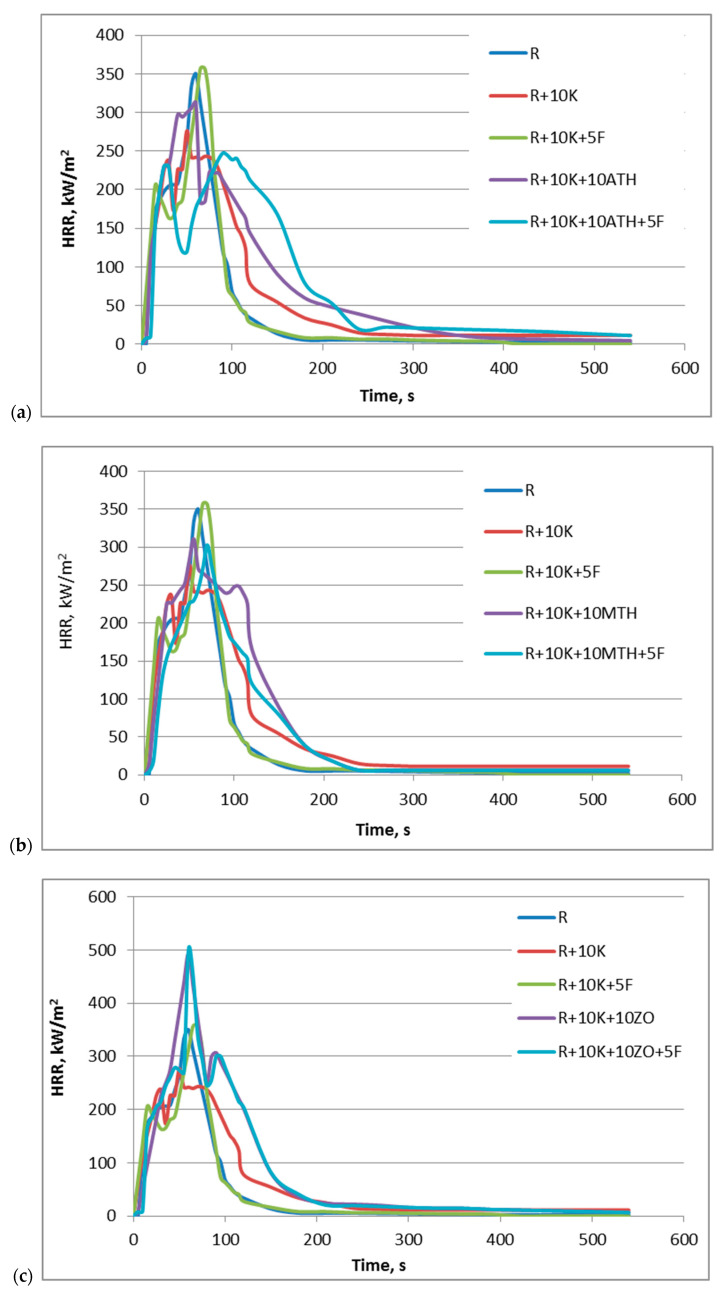
Summary of heat release rate as a function of time for SRPUF samples: (**a**) R; R+10K; R+10K+5F; R+10K+10ATH; R+10K+10ATH+5F, (**b**) R; R+10K; R+10K+5K; R+10K+10MTH; R+10K+10MTH+5F, (**c**) R; R+10K; R+10K+5K; R+10K+10ZO; R+10K+10ZO+5F, (**d**) R; R+10K; R+10K+5F; R+10K+10GE; R+10K+10GE+5F, (**e**) R; R+10K; R+10K+5F; R+10K+10APP; R+10K+10APP+5F, (**f**) R+10K; R+10K+10ATH; R+10K+10MTH; R+10K+10ZO; R+10K+10GE; R+10K+10APP, (**g**) R; R+10K+10ATH+5F; R+10K+10MTH+5F; R+10K+10ZO+5F; R+10K+10GE+5F; R+10K+10APP+5F.

**Figure 8 polymers-12-02943-f008:**
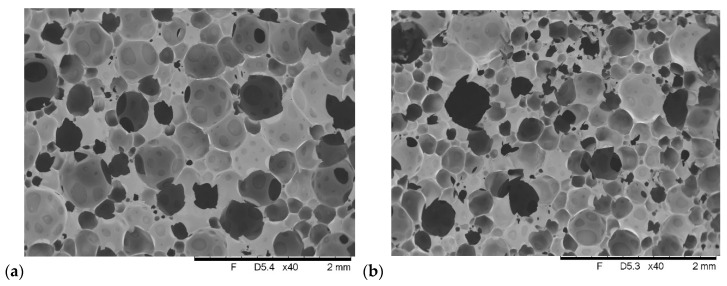
SEM images of selected foams: (**a**) R—reference foam, (**b**) R+10K—reference foam with addition of 10% keratin, (**c**) R+10K+5F—reference foam with addition of 10% keratin and 5% Fyrol, (**d**) R+10K+10GE—reference foam with addition of 10% keratin and 10% Expanding graphite, (**e**) R+10K+10GE+5F—reference foam with addition of 10% keratin, 10% Expanding graphite and 5% Fyrol, (**f**) R+10K+10GE+5F—GE in foam, (**g**) R+10K- keratin fiber in foam.

**Table 1 polymers-12-02943-t001:** The composition of semi-rigid polyurethane foams (SRPUR) and synthesis.

Sample SRPUF	Addition	Amount of Additive % mas/Parts per Hundred Parts of Polyol	Growth Time, s	Gelation Time, s
R*			75	88
R+10K	K	10	63	83
R+10K+5F	K	10	58	90
	F	5		
R+10K+10ATH	K	10	68	93
	ATH	10		
R+10K+10ATH+5F	K	10	61	93
	ATH	10		
	F	5		
R+10 K+10MTH	K	10	63	88
	MTH	10		
R+10K+10MTH+5F	K	10	62	88
	MTH	10		
	F	5		
R+10K+10ZO	K	10	55	74
	ZO	10		
R+10K+10ZO+5F	K	10	49	72
	ZO	10		
	F	5		
R+10 K+10GE	K	10	57	90
	GE	10		
R+10K+10GE+5F	K	10	54	85
	GE	10		
	F	5		
R+10K+10APP	K	10	70	93
	APP	10		
R+10K+10APP+5F	K	10	73	94
	APP	10		
	F	5		

R*—reference foam.

**Table 2 polymers-12-02943-t002:** Summary of thermal analysis results using DSC.

Sample SRPUF	Tg_1_, °C	T, °C	∆H, J/g	Tg_2_, °C	Tg_3_, °C	Tg_4_, °C	D, kg/m^3^
R*	−60.7	80.0	34.5	−63.2	109.2	n.d.	64.6 ± 2.1
R+10K	−61.0	83.2	35.5	−62.5	108.8	n.d.	73.4 ± 0.4
R+10K+5F	−61.8	84.3	41.4	−64.0	107.1	n.d.	80.6 ± 1.5
R+10K+10ATH	−61.4	84.7	31.8	−64.0	108.2	150.0	78.6 ± 1.1
R+10K+10ATH+5F	−61.4	82.2	36.7	−63.8	106.3	148.1	85.2 ± 1.3
R+10K+10MTH	−61.2	80.5	32.5	−62.6	109.6	148.1	76.5 ± 1.3
R+10K+10MTH+5F	−61.6	79.8	27..2	−63.8	107..4	148..1	86.7 ± 1.5
R+10K+10ZO	−62.0	75.3	34.6	−62.8	108.3	148.5	85.0 ± 3.2
R+10K+10ZO+5F	−61.7	80.6	33.4	−63.8	107.6	151.6	87.4 ± 0.1
R+10K+10GE	−61.7	81.9	34.0	−64.6	108.0	n.d.	78.6 ± 0.2
R+10K+10GE+5F	−61.6	81.9	41.0	−64.0	107.1	n.d.	84.1 ± 0.1
R+10K+10APP	−61.4	82.2	36.6	−63.4	108.6	n.d.	77.4 ± 0.3
R+10K+10APP+5F	−60.5	84.7	31.4	−61.8	107.8	n.d.	78.6 ± 1.9

R*—reference foam.

**Table 3 polymers-12-02943-t003:** Summary of TG results obtained during TGA measurements carried out in a nitrogen atmosphere.

Sample SRPUF	T2 %, °C	T5 %, °C	T10 %, °C	T50 %, °C	U_700_, °C
R*	241	261	285	368	16.2
R+10K	232	258	285	380	15.6
R+10K+5F	202	244	276	384	19.0
R+10K+10ATH	202	244	276	384	19.0
R+10K+10ATH+5F	232	257	283	384	17.7
R+10K+10MTH	204	243	275	387	20.9
R+10K+10MTH+5F	236	260	288	383	17.8
R+10K+10ZO	206	252	282	387	21.1
R+10K+10ZO+5F	235	260	289	384	19.2
R+10K+10GE	204	249	281	389	21.4
R+10K+10GE+5F	233	259	285	381	17.9
R+10 K+10APP	227	256	282	367	22.7
R+10K+10APP+5F	203	243	276	369	22.9

R*—reference foam.

**Table 4 polymers-12-02943-t004:** Results of DTG curves analysis obtained by TGA measurements in nitrogen atmosphere.

Sample RPUF	Stage1, °C	T1, °C	V1, %/°C	m1, %	Stage 2, °C	T2, °C	V2, %/°C	m2, %	Stage 3, °C	T3, °C	V3, %/°C	m3, %	Stage4, °C	T4, °C	V4, %/°C	m4, %
R*	n.d.	n.d.	n.d.	n.d.	221–273	262	0.20	7.3	n.d.	n.d.	n.d.	n.d.	273–442	370	0.71	68.3
R+10K	n.d.	n.d.	n.d.	n.d.	222–286	262	0.20	8.7	286–327	314	0.27	10.5	327–453	383	0.99	57.2
R+10K+5F	163–220	205	0.06	2.2	220–283	264	0.17	8.4	283–328	316	0.28	11.2	328–447	386	0.85	52.0
R+10K+10ATH	n.d.	n.d.	n.d.	n.d.	217–288	266	0.20	9.6	288–327	313	0.25	9.1	327–450	385	1.02	56.4
R+10K+10ATH+5F	171–216	204	0.06	1.9	216–297	260	0.17	12.0	297–326	315	0.26	7.3	326–447	388	0.84	50.8
R+10K+10MTH	n.d.	n.d.	n.d.	n.d.	220–286	261	0.18	8.5	286–328	312	0.26	9.6	328–450	384	1.01	56.5
R+10K+10MTH+5F	170–219	200	0.05	1.8	219–280	267	0.17	7.0	280–327	318	0.28	11.0	327–448	388	0.78	51.8
R+10K+10ZO	n.d.	n.d.	n.d.	n.d.	224–284	263	0.18	7.7	284–328	311	0.26	10.1	328–446	383	0.99	54.6
R+10K+10ZO+5F	170–222	203	0.05	2.1	222–278	265	0.16	6.6	279–327	315	0.27	11.2	327–445	391	0.76	51.2
R+10K+10GE	n.d.	n.d.	n.d.	n.d.	222–288	262	0.19	9.2	288–325	314	0.29	8.8	325–451	381	0.91	56.1
R+10K+10GE+5F	188–224	204	0.05	1.8	224–277	262	0.17	6.9	277–325	318	0.26	11.3	325–446	385	0.80	49.8
R+10K+10APP	n.d.	209	0.06	n.d.	217–275	257	0.16	6.9	275–333	317	0.42	19.3	333–445	365	1.02	43.8
R+10K+10APP+5F	170–218	207	0.05	2.1	218–334	317	0.38	25.9	n.d.	n.d.	n.d.	n.d.	334–441	368	1.00	42.7

R*—reference foam.

**Table 5 polymers-12-02943-t005:** Results of TG curves analysis obtained by TGA measurements carried out in air atmosphere.

Sample SRPUF	T2%, °C	T5%, °C	T10%, °C	T50%, °C	U700, %
R*	196	240	246	319	0.06
R+10K	217	239	243	332	0.13
R+10K+5F	219	239	243	338	3.51
R+10K+10ATH	196	232	245	351	3.95
R+10K+10ATH+5F	218	239	242	346	3.58
R+10K+10MTH	197	235	243	351	4.24
R+10K+10MTH+5F	219	239	245	351	5.25
R+10K+10ZO	219	239	242	351	5.25
R+10K+10ZO+5F	198	236	244	365	6.01
R+10K+10GE	219	238	242	342	0.81
R+10K+10GE+5F	198	233	246	358	4.04
R+10K+10APP	215	241	246	393	4.11
R+10K+10APP+5F	195	231	249	394	3.05

R*—reference foam.

**Table 6 polymers-12-02943-t006:** Results of DTG curves analysis obtained by TGA measurements carried out in air atmosphere.

Sample SRPUF	Stage1, °C	T1, °C	V1, %/°C	m1, %	Stage 2, °C	T2, °C	V2, %/°C	m2, %	Stage 3, °C	T3, °C	V3, %/°C	m3, %	Stage4, °C	T4, °C	V4, %/°C	m4, %
R*	226–306	248	1.81	44.6	n.d.	n.d.	n.d.	n.d.	306–379	338	0.16	9.8	374–663	520	0.48	42.4
R+10K	232–267	244	1.92	30.4	267–306	278	0.40	12.6	306–394	328	0.18	11.3	394–636	514	0.52	43.3
R+10K+5F	235–256	246	1.84	19.4	256–381	280	0.47	28.1	n.d.	n.d.	n.d.	n.d.	381–660	514	0.49	46.0
R+10K+10ATH	218–270	244	1.66	30.7	270–312	279	0.41	13.0	312–399	332	0.16	10.1	399–653	516	0.48	40.8
R+10K+10ATH+5F	223–263	246	1.70	23.0	263–383	280	0.48	25.2	n.d.	n.d.	n.d.	n.d.	383–699	514	0.46	43.8
R+10K+10MTH	222–269	245	1.62	29.0	269–304	280	0.37	14.0	304–408	366	0.12	14.0	408–663	513	0.46	43.4
R+10K+10MTH+F	221–265	245	1.81	25.1	265–340	280	0.43	20.0	340–400	368	0.11	5.9	400–668	513	0.44	41.1
R+10K+10ZO	227–266	243	1.71	27.6	266–310	278	0.38	13.4	310–391	329	0.16	9.8	391–650	511	0.54	41.4
R+10 K+10 ZO+5F	228–261	245	1.81	21.8	261–381	278	0.42	25.4	n.d.	n.d.	n.d.	n.d.	381–638	510	0.48	42.8
R+10K+10GE	222–268	245	1.67	28.7	268–394	276	0.43	23.8	n.d.	n.d.	n.d.	n.d.	394–630	519	0.49	41.7
R+10K+10GE+5F	227–265	248	1.36	21.1	265–382	284	0.52	26.4	n.d.	n.d.	n.d.	n.d.	382–660	520	0.46	44.0
R+10 K+10APP	229–263	247	1.24	19.3	263–381	277	0.48	26.8	n.d.	n.d.	n.d.	n.d.	381–622	517	0.46	42.6
R+10K+10APP+5F	231–267	252	0.94	17.8	267–385	285	0.52	26.6	n.d.	n.d.	n.d.	n.d.	385–612	515	0.45	40.7

R*—reference foam.

**Table 7 polymers-12-02943-t007:** Summary of the results obtained on the basis of the flammability analysis carried out with the use of a cone calorimeter.

Sample SRPUF	TTI, s	TTF, s	pHRRm, kW/m^2^	MARHE, kW/m^2^	pHRR/ t(pHRR), kW/m^2^s	THR, MJ/m^2^	MLR, g/s	SEA, m^2^/kg	TSP, m^2^/m^2^	pCO, kg/kg	pCO_2_, kg/kg
R*	4	154	359	217	6	25	0.14	2298	502	0.31	15.31
R+10K	2	218	283	213	5	31	0.23	640	612	0.35	15.54
R+10K+5F	2	166	350	210	6	26	0.20	458	608	0.59	12.97
R+10K+10ATH	4	216	307	195	6	31	0.27	365	460	0.29	14.81
R+10K+10ATH+5F	12	322	248	177	3	39	0.24	1062	1102	0.47	13.06
R+10K+10MTH	4	226	301	215	6	35	0.23	805	1002	0.21	13.90
R+10K+10MTH+5F	14	260	298	169	4	29	0.26	697	973	0.32	12.41
R+10K+10ZO	6	196	491	251	8	40	0.32	688	975	0.32	14.00
R+10K+10ZO+5F	8	218	453	258	7	38	1.84	1295	987	0.41	14.67
R+10K+10GE	6	490	165	95	10	32	0.16	134	194	0.44	16.21
R+10K+10GE+5F	8	436	147	97	8	29	0.15	813	687	0.31	12.68
R+10K+10APP	4	170	341	235	4	30	0.28	757	1071	0.34	12.54
R+10K+10APP+5F	2	164	305	240	4	30	3.86	278	297	0.27	12.42

R*—reference foam.

**Table 8 polymers-12-02943-t008:** Summary of the results of determining the dimensional stability of foams and water absorption.

Sample SRPUF	Dimensional Stability %			Water Absorption, %
	*X*-Axis (Opposite to the Growth Direction)	*Y*-Axis (Opposite to the Growth Direction)	*Z*-Axis (According to the Growth Direction)	
R*	0.77 ± 0.42	0.73 ± 0.25	4.97 ± 1.62	8.78 ± 0.64
R+10K	0.60 ± 0.11	0.46 ± 0.21	0.86 ± 1.85	7.95 ± 1.06
R+10K+5F	1.29 ± 0.54	1.28 ± 0.79	4.11 ± 1.05	15.18 ± 4.34
R+10K+10ATH	1.09 ± 0.22	0.58 ± 0.19	4.76 ± 2.54	12.93 ± 1.11
R+10K+10ATH+5F	0.99 ± 0.31	1.80 ± 0.04	6.74 ± 4.71	13.19 ± 3.45
R+10K+10MTH	0.92 ± 0.02	1.50 ± 0.39	1.19 ± 0.30	16.17 ± 1.34
R+10K+10MTH+5F	−0.81 ± 0.31	3.24 ± 1.38	2.66 ± 0.98	20.51 ± 3.65
R+10K+10ZO	−1.26 ± 0.26	2.53 ± 0.48	4.51 ± 0.90	11.67 ± 1.66
R+10K+10ZO+5F	0.75 ± 0.19	1.74 ± 1.47	3.11 ± 0.19	14.98 ± 0.81
R+10K+10GE	−6.84 ± 2.93	8.13 ± 3.02	3.15 ± 0.36	9.95 ± 1.25
R+10K+10GE+5F	0.98 ± 0.08	1.01 ± 0.19	2.71 ± 0.70	12.22 ± 1.14
R+10K+10APP	−1.88 ± 0.46	4.67 ± 3.52	3.04 ± 1.06	10.48 ± 2.89
R+10K+10APP+5F	1.21 ± 0.16	1.26 ± 0.88	3.83 ± 1.80	12.66 ± 2.89

R*—reference foam.

**Table 9 polymers-12-02943-t009:** Characteristic of the foam and composites microstructure.

Sample	d [um]	AR
R	25.01 ± 15.90	1.30 ± 0.20
R+10K	23.80 ± 14.08	1.25 ± 0.21
R+10K+5F	23.71 ± 15.77	1.27 ± 0.20
R+10K+10GE	24.84 ± 14.30	1.32 ± 0.26
R+10K+10GE+5F	24.58 ± 17.21	1.25 ± 0.17

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
