# Peer review of "Composites of Semi-Rigid Polyurethane Foams with Keratin Fibers Derived from Poultry Feathers and Flame Retardant Additives"

_polymers, 2020, doi:10.3390/polym12122943_

Round 1
Reviewer 1 Report
This work reported modified semi-rigid composites of polyurethane foams (SRPUF) by the addition of keratin flour from poultry feathers and flame retardant additives. 10% by mass of keratin fibres was added to the foams as well as halogen-free flame retardant additives such as: Fyrol PNX, expandable graphite, metal oxides, in such amount that their total mass did not exceed 15%. The addition of the filler limits the amount of smoke generated during foam burning and achieves favourable reduction of heat and smoke release rate with the addition of 10% keratin fibres and 10% expandable graphite. Overall, the manuscript is well written and organized. However, there are still some issues which remain to be addressed.
- What about the mechanical properties of modified foams?
- Flame retardant mechanisms are still not clear. Please provide an in-depth understanding of flame retardant mechanisms by investigating char residues in the condensed phase using FTIR, Raman, XRD….Please refer to the following published papers. (Polymer, 153 (2018) 616-625; Journal of Applied Polymer Science, 111 (2009) 1115-1143)
- Please provide more detailed description regarding cone calorimeter, and some important papers are missing (Journal of Hazardous Materials, 401 (2021) 123342; Journal of Materials Chemistry A, 4 (2016) 7330-7340).
Author Response
- What about the mechanical properties of modified foams?
Unfortunately, we have not done such research. Currently, The Metrology Laboratory has closed due to COVID-19 until further notice. Therefore, we cannot perform mechanical tests at present.
- Flame retardant mechanisms are still not clear. Please provide an in-depth understanding of flame retardant mechanisms by investigating char residues in the condensed phase using FTIR, Raman, XRD….Please refer to the following published papers. (Polymer, 153 (2018) 616-625; Journal of Applied Polymer Science, 111 (2009) 1115-1143)
We thank the Reviewer for the valuable comment; however, due to the limited review time and the lack of char, the proposed analysis was not included in the article. Performing burning tests and after that investigating char residues using FTIR, Raman, XRD studies, although extremely valuable, is not possible in such a short time. Based on the proposed as well as other articles, the authors extended the analysis and added comments regarding the mechanism of action of the fire retardants used. We appreciate the Reviewer's remarks and will apply it in future articles.
- Please provide more detailed description regarding cone calorimeter, and some important papers are missing (Journal of Hazardous Materials, 401 (2021) 123342; Journal of Materials Chemistry A, 4 (2016) 7330-7340).
We agree with the recommendation and based on the provided article, carefully improved that inaccuracy in the manuscript.
Reviewer 2 Report
1) Pyrene is known to be a compound with four fused aromatic ring. Please define what is "anti-pyrene"?
2) Keratin is hydrophobic. Can feather be dispersed well in polyol? Can the author disclose the elemental composition of the feather?
3)Page 2- Line 76: change melamine formula to (C3N3(NH2)3).
Author Response
- Pyrene is known to be a compound with four fused aromatic ring. Please define what is "anti-pyrene"?
The authors wish to thank the Reviewer for drawing attention to the error that was made. "Anti-pyrene" have been replaced by “fire retardants”.
- Keratin is hydrophobic. Can feather be dispersed well in polyol? Can the author disclose the elemental composition of the feather?
Chicken feathers were characterised by the following contents: sulphur content of 2.9%, nitrogen content of 15.5%, and ash content of about 1%.
3. Page 2- Line 76: change melamine formula to (C3N3(NH2)3).
Changed
Round 2
Reviewer 1 Report
The authors has addressed all the issues raised by reviewers well. I recommend the accepance of the revised manuscript in the current form.
This manuscript is a resubmission of an earlier submission. The following is a list of the peer review reports and author responses from that submission.
Round 1
Reviewer 1 Report
The manuscript describes the synthesis of polyurethane foams containing various filler materials and evaluates their effects on composition, mechanical properties, and flame retardant behavior of the foams. The work presents an interesting approach to use poultry feather derived materials and performs thorough characterization of the materials. However, the manuscript is poorly written and the figures are unclear and/or too busy to effectively communicate the research to the reader. Throughout the manuscript it is very apparent that there were multiple writers, each with a distinct voice. The entire manuscript needs to be revised to provide a continuous voice. Significantly, the sample naming convention is not clearly described and thus it was very difficult to comprehend much of the discussion throughout the manuscript. A thorough proofread and corrections by native English language speaker is required prior to resubmission for publication. Gel fraction analysis is highly recommended on the foams to evaluate effect of additives on polymerization of polyurethanes, particularly since there was significant residual isocyanate following polymerization (as seen at 2200 cm-1 in the IR). For these reasons, this manuscript is not recommended for publication. Some specific recommendations are listed below:
abstract
-SRPUF – rigid or semi-rigid? Abstract states rigid polyurethane foam, however the acronym SRPUF does not align
-flame retardant additives – In the abstract, list the additives used in the study. Important for prospective readers to know what to expect – at least the classes of compounds, e.g. metal oxides, etc.
-thorough proofread for English language of abstract
Introduction
-revise structure of introduction - it begins with many short 1 or 2 sentence paragraphs. then ends with very long paragraph
--the first 7 paragraphs need to be significantly improved, perhaps combined into one or two, much improved paragraphs
-There are clearly multiple authors – rework introduction so that it exhibits a single, unified voice
-antipyrene vs anti-pyrene – be consistent with usage
-second to last, very long paragraph – break up into at least 2 paragraphs
Materials and methods
-changed bulleted list to sentences in paragraphs. Also, be consistent with denoting manufacturer – denote manufacturer for all chemicals, and also be consistent (i.e. all in parenthesis)
-page 4, line 152 – “mixed with baguette” – what is this? Needs correction.
-table 1 – caption list SRPUF, yet column 1 header lists RPUF. Which one? Be consistent throughout the rest of the manuscript. It is unclear what the subject material of the manuscript is, RPUF or SRPUF.
-table 1 – the component abbreviations are unclear. This could be corrected by an improved mateirals and methods section (see comment above). For example, what is R? Use normal conventions by listing the abbreviations for a name after the words is first used in the manuscript in parenthesis
-page 5, lines 159-160 – provide abbreviations for infrared spectroscopy, scanning electron microscopy, etc.
-2.2.1 – how was the volume measured? What is simply the geometric dimensions? Therefore the density takes into account the numerous voids and bubbles? Please clarify in the text
-page 6, equation 2 – correct the symbol for thermal conductivity coefficient
-Page 6, line 224 – begins with discussion of SRPUR, however the prior discussion and table discussed RPUF. Correct and be consistent
-figure 1 – increase the resolution of figure.
-figure 1- the peak at 1709 is mislabed as C-O. correct to C=O for the urethane carbonyl
-figure 1 – significant residual unreacted isocyanate remained in the samples. This needs to be addressed. Improved curing by placing in oven ~60 C for multiple hours to several days, or wash with solved (THF) to remove unreacted fragments.
-recommend running gel fraction analysis to evaluate additive and filler material and % cure.
-page 7, line 266 – “This comparison indicates that the additives used cause the slightly changes the degree of phase separation of these foams”. This conclusion appears unsupported by the IR data. The authors need to expand discussion to support this claim.
-figure 3 – needs significant improvement. Make each plot a different color, currently there are 2 green lines. Also, there is too much overlayed data and arrows on the plot. Label each curve – heating 1, cooling, heating 3? It is not clear
Page 9, lines 298-316 – change division signs to dashes to indicate range in temperatures
Reviewer 2 Report
1) Expandable Graphite not Expanded Graphite should be used throughout the paper.
2) What is the particle size and Starting Expansion Temperature of EG?
3) Could the authors display the SEM picture of the keratin fiber. What is the aspect ratio of the fiber?
4) What are the particle sizes of the ATH, MDH, and ZnO?
5) For Fyrol PNX, "oligomeric non-reactive phosphate ester" shoud be added.
6) In the Introduction, rigid PU foam was discussed extensively. Why the paper is focused on semi-rigid PU foam?
7) For comparison of FR performance, it would be nice to report the performance of R+10GE.
Reviewer 3 Report
This work reported rigid composites of polyurethane foams (SRPUF) modified with the addition of keratin flour from poultry feathers and flame retardant additives were manufactured. Thermal and mechanical properties were tested, water absorption, dimensional stability, apparent density and flammability of produced foams were determined. The addition of a filler made of keratin fibers significantly limits the amount of smoke and heat generated during foam burning. This manuscript presents a comprehensive and interesting work. However, some issues need to be addressed before publication in Polymers.
- Many critical papers regarding flame retardant rigid polyurethane foams are missing in the introduction. (Composites Part a-Applied Science and Manufacturing, 112 (2018) 142-154; Industrial & Engineering Chemistry Research, 55 (2016) 10813-10822)
- Please further characterize the morphology and thermal stability of Keratin Fibers by using SEM, and TGA.
- Please investigate the interfacial interaction and morphology of SRPUF systems by using SEM.
- How many specimens are repeated for cone calorimetry tests? Please provide the experimental errors for TTI, TTF, pHRR…
- The detailed investigation on flame retardant mechanisms of SRPUF systems. It is suggested to add the morphology and structure analysis of char residues from the selected SRPUF samples by using FTIR, Raman and SEM. The authors can refer to the following papers. (Journal of Hazardous Materials, 401 (2021) 123342; Chemical Engineering Journal, (2020) 125829; Journal of Materials Chemistry A, 4 (2016) 7330-7340)